# DEFENDING MEMBERSHIP INFERENCE ATTACKS VIA PRIVACY-AWARE SPARSITY TUNING

## ABSTRACT

Over-parameterized models are typically vulnerable to membership inference attacks, which aim to determine whether a specific sample is included in the training of a given model. Previous Weight regularizations (e.g., $\ell_1$ regularization) typically impose uniform penalties on all parameters, leading to a suboptimal tradeoff between model utility and privacy. In this work, we first show that only a small fraction of parameters substantially impact the privacy risk. In light of this, we propose Privacy-aware Sparsity Tuning (**PAST**)—a simple fix to the $\ell 1$ Regularization—by employing adaptive penalties to different parameters. Our key idea behind PAST is to promote sparsity in parameters that significantly contribute to privacy leakage. In particular, we construct the adaptive weight for each parameter based on its privacy sensitivity, i.e., the gradient of the loss gap with respect to the parameter. Using PAST, the network shrinks the loss gap between members and non-members, leading to strong resistance to privacy attacks. Extensive experiments demonstrate the superiority of PAST, achieving a state-of-the-art balance in the privacy-utility trade-off.

## 1 INTRODUCTION

Modern neural networks are trained in an over-parameterized regime where the parameters of the model exceed the size of the training set (Zhang et al., 2021). While the huge amount of parameters empowers the models to achieve impressive performance across various tasks, the strong capacity also makes them particularly vulnerable to membership inference attacks (MIAs) (Shokri et al., 2017). In MIAs, attackers aim to detect if a sample is utilized in the training of a target model. Membership inference can cause security and privacy concerns in cases where the target model is trained on sensitive information, like health care (Paul et al., 2021), financial service (Mahalle et al., 2018), and DNA sequence analysis (Arshad et al., 2021). Therefore, it is of great importance to design robust training algorithms for over-parameterized models to defend against MIAs.

The main challenge of protecting against MIAs stems from the extensive number of model parameters, allowing to easily disclose the information of training data (Tan et al., 2022a). Therefore, previous works reduce the over-complexity of neural networks by weight regularization, like $\ell_1$ or $\ell_2$ regularization. These regularization techniques impose uniform penalties on all parameters with large values, reducing the overfitting to the training data. However, if not all parameters contribute equally to the risk of leaking sensitive information, the uniform penalties can lead to a suboptimal tradeoff between model utility and privacy. The question is:

*Are all parameters equally important in terms of privacy risk?*

In this work, we answer this question by an empirical analysis of parameter sensitivity in terms of privacy risk. In particular, we take the loss gap between member and non-member examples as a proxy for privacy risk and compute its gradient with respect to each model parameter. We find that only a small fraction of parameters substantially impact the privacy risk, whereas the majority have little effect. Thus, applying uniform penalties to all parameters is inefficient to defend against MIAs and may unnecessarily restrict the model's capacity.

To address this issue, we propose Privacy-Aware Sparsity Tuning (PAST), a simple fix to $\ell_1$ regularization that employs adaptive penalties to different parameters in a deep neural network. The

key idea behind PAST is to promote sparsity in parameters that significantly contribute to privacy leakage. In particular, we modulate the intensity of $\ell_1$ regularization for model parameters based on their privacy sensitivity, i.e., the gradient of the loss gap with respect to the parameters. In effect, our method not only stringently regularizes sensitive parameters, but also maintains the model utility by sparing less sensitive parameters from excessive regularization. Trained with the proposed regularization, the network shrinks the loss gap between members and non-members, leading to strong resistance to privacy attacks.

To verify the effectiveness of our method, we conduct extensive evaluations on five datasets, including Texas100 (Texas Department of State Health Services, 2006), Purchase100 (Kaggle, 2014), CIFAR-10/100 (Krizhevsky et al., 2009), and ImageNet (Russakovsky et al., 2015) datasets. The results demonstrate our methods can improve utility-privacy trade-offs across a variety of attacks based on neural networks, metrics, and data augmentation. For example, our method significantly diminishes the attack advantage of loss attack from 14.8% to 5.2% - a relative reduction of 64.9% in privacy risk, whilst preserving the test accuracy.

Our contributions are summarized as follows:

- We empirically analyze the importance of model parameters in privacy risk. We show that only a few parameters substantially impact the privacy risk, whereas the majority have little effect. This suggests that the MIA defense can focus on a few important parameters.

- We introduce PAST – a simple and effective regularization method, which promotes sparsity in parameters that significantly contribute to privacy leakage. We show that PAST can effectively improve the utility-privacy trade-offs across a variety of attacks.

- We perform ablation studies that lead to an improved understanding of our method. In particular, we contrast with alternative methods (e.g., $\ell 1$ or $\ell 2$ regularization) and demonstrate the advantages of PAST. We hope that our insights inspire future research to further explore weight regularization for MIA defense.

## 2 PRELIMINARIES

### 2.1 BACKGROUND

**Setup**  In this paper, we study the problem of membership inference attacks in $K$-class classification tasks. Let the feature space be $\mathcal{X} \subset \mathbb{R}^d$ and the label space be $\mathcal{Y} = \{1, \ldots, K\}$. Let us denote by $(\boldsymbol{x}, y) \in (\mathcal{X} \times \mathcal{Y})$ an example containing an instance $\boldsymbol{x}$ and a real-valued label $y$. Given a training dataset $\mathcal{S} = \{(\boldsymbol{x_n}, y_n)\}_{i=1}^N$ *i.i.d.* sampled from the data distribution $\mathcal{P}$, our goal is to learn a model $h_\theta$ with trainable parameters $\theta \in \mathbb{R}^p$, that minimizes the following expected risk:

$$\mathbb{R}(h_\theta) = \mathbb{E}_{(\boldsymbol{x}, y) \sim \mathcal{P}}[\mathcal{L}(h_\theta(\boldsymbol{x}), y)] \tag{1}$$

where $\mathbb{E}_{(\boldsymbol{x}, y) \sim \mathcal{P}}$ denotes the expectation over the data distribution $\mathcal{P}$ and $\mathcal{L}$ is a conventional loss function (such as cross-entropy loss) for classification. In modern deep learning, the neural network $h_\theta$ is typically over-parameterized, allowing to easily disclose the information of training data (Tan et al., 2022a).

**Membership Inference Attacks**  Given a data point $(\boldsymbol{x}, y)$ and a trained target model $h_\mathcal{S}$, attackers aim to identify if $(\boldsymbol{x}, y)$ is one of the members in the training set $\mathcal{S}$, which is called membership inference attacks (MIAs) (Shokri et al., 2017; Yeom et al., 2018; Salem et al., 2019). In MIAs, it is generally assumed that attackers can query the model predictions $h_\theta(\boldsymbol{x})$ for any instance $\boldsymbol{x}$. Here, we focus on standard black-box attacks (Irolla & Châtel, 2019), where attackers can access the knowledge of model architecture and the data distribution $\mathcal{P}$.

In the process of attack, the attacker has access to a query set $Q = \{(\boldsymbol{z}_i, m_i)\}_{i=1}^J$, where $\boldsymbol{z}_i$ denotes the $i$-th data point $(\boldsymbol{x}_i, y_i)$ and $m_i$ is the membership attribute of the given data point $(\boldsymbol{x}_i, y_i)$ in the training dataset $\mathcal{S}$, i.e., $m_i = \mathbb{I}[(\boldsymbol{x}_i, y_i) \in \mathcal{S}]$. In particular, the query set $Q$ contains both member (training) and non-member samples, drawn from the data distribution $\mathcal{P}$. Then, the attacker $\mathcal{A}$ can be formulated as a binary classifier, which predicts $m_i \in \{0, 1\}$ for a given example $(\boldsymbol{x}_i, y_i)$ and a target model $h_\theta$: $\mathcal{A}(\boldsymbol{x}_i, y_i; h_\theta) \rightarrow \{0, 1\}$.

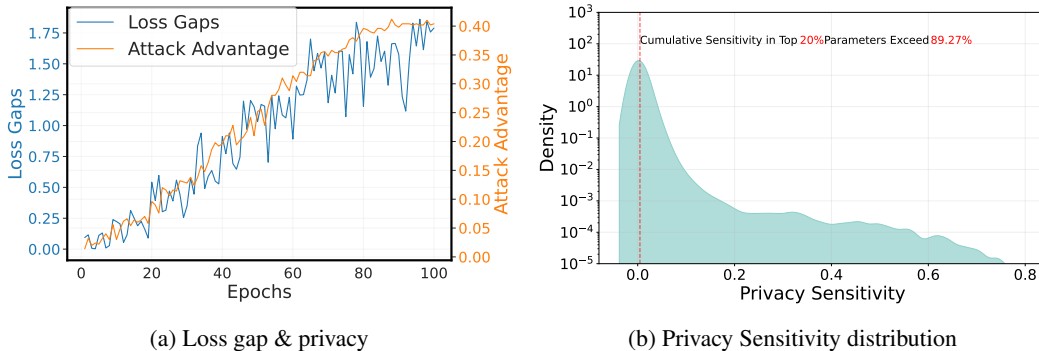

(a) Loss gap & privacy

(b) Privacy Sensitivity distribution

Figure 1: (a) Loss gaps and attack advantage during standard training. The attack advantage increases synchronously with the loss gap during the training process, showing the privacy leakage of over-parameterization, and thus we consider the loss gap as a proxy; (b) The privacy sensitivity distribution across parameters. Only a small fraction of parameters substantially impacts the privacy risk. (The cumulative sensitivity in the top 20% parameters exceeds 89.27% of the total.)

**Weight regularization** The privacy risk of deep neural networks is often associated with their over-parameterized nature. Intuitively, the huge amount of parameters enables the model to encapsulate extensive information of the training data, potentially leading to unintended privacy leakages. Previous work shows theoretically that increasing the number of model parameters renders them more vulnerable to membership inference attacks (Tan et al., 2022b). To address this issue, weight regularization is typically employed to alleviate the membership inference, such as $\ell_1$ and $\ell_2$ regularizations (Hoerl & Kennard, 1970; Tibshirani, 1996; Schmidt et al., 2007). Formally, the weight regularization can be formalized as:

$$\mathbb{R}_{\text{reg}}(h_\theta) = \mathbb{E}_{(\boldsymbol{x},y)\sim\mathcal{P}}[\mathcal{L}(h_\theta(\boldsymbol{x}),y)] + \lambda R(h_\theta) \tag{2}$$

where $\mathcal{L}(\cdot)$ denotes the classification loss, $\lambda$ is the hyperparameter that controls the importance of the regularization term, and $R(\theta)$ is typically chosen to impose a penalty on the complexity of $h_\theta$. For example, $\ell_1$ and $\ell_2$ regularizations penalize the norm of model parameters as follows: $R(h_\theta) = \|\theta\|_r^r$, where $r$ denotes the order of the norm.

Previous work (Tan et al., 2022b) shows that one can reduce vulnerability to MIAs by reducing the number of effective parameters, such as utilizing the sparsification effect of $\ell_1$ regularization. However, this comes at the cost of inferior generalization performance (utility) due to the "double descent" effect (Belkin et al., 2019; 2020; Dar et al., 2021), wherein generalization error decreases with increased overparameterization. This challenge stems from the uniform penalty applied to all parameters, ignoring their potentially varying importance in terms of privacy leakage.

## 3 METHOD: PRIVACY-AWARE SPARSITY TUNING

In this section, we start by analyzing the privacy sensitivity of model parameters and find that most parameters contribute only marginally to the privacy risk. Subsequently, we design a weighted $\ell_1$ regularization that takes into account the privacy sensitivity of each parameter.

### 3.1 MOTIVATION

In this part, we aim to figure out whether the model parameters are equally important in terms of privacy risk. In particular, we perform standard training with ResNet-18 (He et al., 2016) on CIFAR-10 (Krizhevsky et al., 2009). We train the models using SGD with a momentum of 0.9, a weight decay of 0.0005, and a batch size of 128. We set the initial learning rate to 0.01 and decrease it using a cosine scheduler (Loshchilov & Hutter, 2017) throughout the training. In the analysis, we construct the datasets of members $\mathcal{S}_m$ and nonmembers $\mathcal{S}_n$ by randomly sampling two subsets with 10000 examples each from the training set and the test set, respectively.

**Loss gaps as proxy of privacy risk** In this study, we use the loss gap between member and non-member examples as a proxy for privacy risk: $\mathcal{G}(\mathcal{S}_m, \mathcal{S}_n; h_\theta) = \left| \frac{1}{|\mathcal{S}_m|} \sum_{(\boldsymbol{x},y) \in \mathcal{S}_m} \mathcal{L}(h_\theta(\boldsymbol{x}), y) - \frac{1}{|\mathcal{S}_n|} \sum_{(\boldsymbol{x},y) \in \mathcal{S}_n} \mathcal{L}(h_\theta(\boldsymbol{x}), y) \right|$. Specifically, we calculate the difference between the average losses of members and non-members. A larger loss gap indicates a higher privacy risk, as it suggests the model is more susceptible to membership inference attacks (MIAs). It has been shown that the loss function can determine the optimal attacks in membership inference (Sablayrolles et al., 2019). As demonstrated in Figure 1a, the model training increases both the attack advantage (See the definition in Section 4.1) and the loss gap simultaneously, supporting the use of the loss gap as a proxy for the privacy risk.

**Most parameters contribute only marginally to the privacy risk** We measure the privacy sensitivity of a parameter $\theta_i$ by the gradient of the loss gap with respect to the parameter: $\nabla_{\theta_i}(\mathcal{G}(\mathcal{S}_m, \mathcal{S}_n; h_\theta))$. Figure 1b illustrates the privacy sensitivity distribution of model parameters. The results show that only a small fraction of parameters substantially impact the privacy risk, whereas the majority have little effect. For example, 97% of the parameters have privacy sensitivities lower than 0.1. The cumulative sensitivity in the top 20% parameters exceeds 89.27% of the total. These findings suggest that applying uniform penalties to all parameters is inefficient to defend against MIAs and may unnecessarily restrict the model's capacity. Instead, the weight regularization can be more efficiently applied by focusing on the most sensitive parameters rather than the entire parameter set. We proceed by introducing our method, targeting this problem.

## 3.2 METHOD

In the previous analysis, we demonstrate that the privacy risk can be alleviated by reducing the number of effective parameters with weight regularization techniques. Moreover, we show that most parameters contribute only marginally to the privacy risk, suggesting that the weight regularization can be focused on the most sensitive parameters. Thus, our key idea is to promote sparsity specifically within the subset of parameters that significantly contribute to privacy leakage.

**Privacy-Aware Sparsity Tuning** In this work, we introduce Privacy-Aware Sparsity Tuning (dubbed **PAST**), a simple fix to $\ell_1$ regularization that employs adaptive penalties to different parameters in a deep neural network. Formally, the objective function of PAST is given by:

$$
\begin{aligned}
\mathbb{R}_{\text{PAST}}(h_\theta) &= \mathbb{E}_{(\boldsymbol{x},y) \sim \mathcal{P}}[\mathcal{L}(h_\theta(\boldsymbol{x}), y)] + \lambda R(h_\theta) \\
&= \mathbb{E}_{(\boldsymbol{x},y) \sim \mathcal{P}}[\mathcal{L}(h_\theta(\boldsymbol{x}), y)] + \lambda \sum_i \gamma_i |\theta_i|
\end{aligned}
\tag{3}
$$

where $\lambda$ is the hyperparameter that controls the importance of the regularization term and $\gamma_i$ denotes the adaptive weight of the parameter $\theta_i$. We expect larger weights for those parameters with higher privacy sensitivity, and smaller weights for those with lower sensitivity. Using the $\ell_1$ norm, the regularization can encourage those sensitive parameters to be zero, thereby improving the defense performance against MIAs.

In particular, we modulate the intensity of $\ell_1$ regularization for model parameters based on their privacy sensitivity, i.e., the gradient of the loss gap with respect to the parameters. Let $\mathcal{S}_m$ and $\mathcal{S}_n$ denote the subsets of members and non-members, respectively. For notation shorthand, we use $\mathcal{G}_\theta$ to denote the loss gap $\mathcal{G}(\mathcal{S}_m, \mathcal{S}_n; h_\theta)$ of the model $h_\theta$ on $\mathcal{S}_m$ and $\mathcal{S}_n$. Then, we compute the normalized privacy sensitivity of each parameter $\theta_i$ in its associated module (e.g., linear layer):

$$
\gamma_i = \frac{|\mathcal{M}(\theta_i)| \nabla_{\theta_i} \mathcal{G}_\theta}{\sum_{\theta_j \in \mathcal{M}(\theta_i)} \nabla_{\theta_j} \mathcal{G}_\theta},
$$

where $\mathcal{M}(\theta_i)$ denotes the associated module of the parameter $\theta_i$. Equipped with the adaptive weight, the final regularization of PAST is :

$$
R(h_\theta) = \sum_i \gamma_i^\alpha |\theta_i|,
\tag{4}
$$

where $\alpha$ is the focusing parameter that adjusts the rate at which sensitive parameters are up-weighted. When $\alpha = 0$, the regularization is equivalent to the standard $\ell_1$ regularization. As $\alpha$

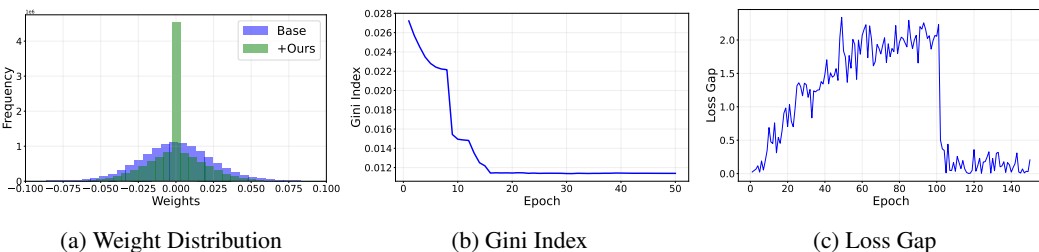

(a) Weight Distribution       (b) Gini Index       (c) Loss Gap

Figure 2: (a) Weight distribution before (Base) and after regularization (Ours). Weights of Ours is clearly more concentrated around 0 and thus is sparser compared to the base; (b) Gini index (criterion for sparsity) during the regularization process. The Gini index continues decreasing during tuning, which also demonstrates the sparsity effect of PAST; (c) Loss gap throughout the whole training process. The regularization (beginning at epoch 100) quickly reduces the loss gap, leading to strong resistance to privacy attacks.

increases, the regularization puts more focus on the few parameters with high privacy sensitivity. The adaptive weight enables to relax the penalties for insensitive parameters while imposing stricter penalties on sensitive parameters. Note that the $\gamma_i$ does not require a gradient in backpropagation, so it is detached from the computational graph, leading to efficient implementation of PAST.

**Implementation of tuning** Standard $\ell_1$ regularization is usually employed from the beginning of model training to alleviate the overfitting. This makes it challenging to achieve a good tradeoff between privacy and utility, as a strict regularization degrades the model's capacity for generalization. Instead, we propose to employ the regularization after the model convergence in the training with the classification loss. In particular, we first train the model using the loss (Equation (3)) with $\lambda = 0$ until convergence. Then, we increase the value of $\lambda$ to tune the model with the regularized loss. The tuning mechanism allows our method to be compatible with trained models, instead of requiring retraining from scratch.

By applying our method during tuning, we not only stringently regularize sensitive parameters but also preserve model utility by sparing less sensitive parameters from excessive regularization. Specifically, Figures 2a and 2b confirm the effectiveness of our method in mitigating over-parameterization both intuitively and quantitatively: after PAST tuning, more parameters are concentrated around 0 in the weight distribution, and the Gini index—a measure of sparsity—also significantly decreases. Figure 2c further illustrates the impact of our tuning on the loss gap, which sharply declines after the beginning of tuning at epoch 100, demonstrating the method's ability to quickly reduce the loss gap and enhance resistance to privacy attacks.

## 4 EXPERIMENTS

### 4.1 SETUP

**Datasets** In our evaluation, we employ five datasets: Texas100 (Texas Department of State Health Services, 2006), Purchase100 (Kaggle, 2014), CIFAR-10, CIFAR-100 (Krizhevsky et al., 2009), and ImageNet (Russakovsky et al., 2015). Following previous work (Liu et al., 2024b), we split each dataset into six subsets, with each subset alternately serving as the training, testing, or inference set for the target and shadow models. The inference set was used by our method and adversarial training algorithms that incorporate adversary loss—such as Mixup+MMD (Li et al., 2021) and adversarial regularization (Nasr et al., 2018). In our method, the inference set was used to obtain the comparison information between members and non-members.

**Training details** We train the models using SGD with a momentum of 0.9, a weight decay of 0.0005, and a batch size of 128. We set the initial learning rate to 0.01 and drop it using a cosine scheduler (Loshchilov & Hutter, 2017) with $T_{max} = epochs$. For CIFAR-10, we conduct training using an 18-layer ResNet (He et al., 2016), with 100 epochs of standard training and 50 epochs of sparse tuning. In the case of ImageNet and CIFAR-100, we employ a 121-layer DenseNet (Huang

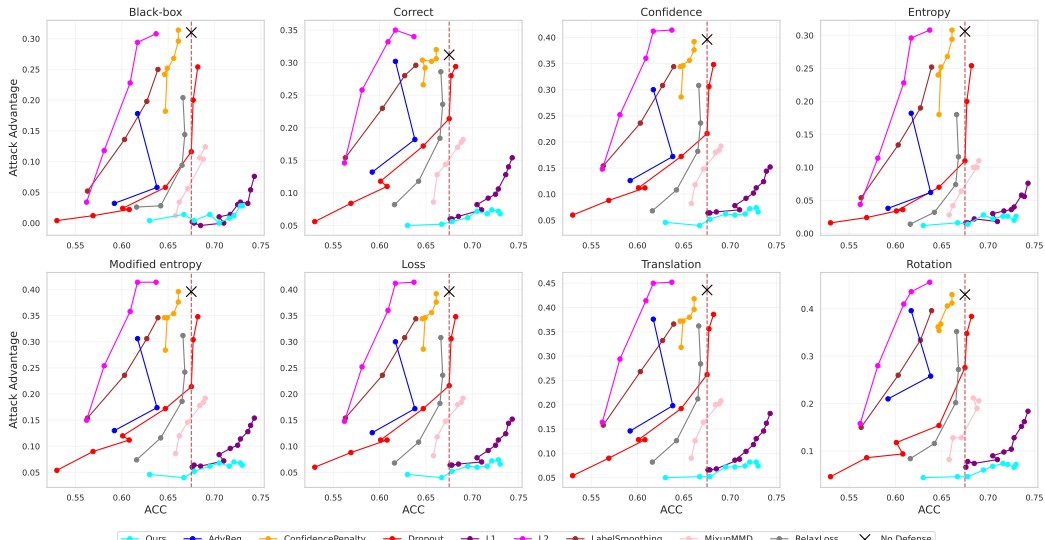

Figure 3: Comparisons of five defense mechanisms on CIFAR-10 dataset utilizing Resnet18 architecture. Each subplot is allocated to a distinct attack method, wherein individual curves represent the performance of a defense mechanism under different hyperparameter settings. The horizontal axis represents the target models' test accuracy (the higher the better), and the vertical axis represents the corresponding attack advantage (defined in Definition 5, the lower the better). To underscore the disparity between the defense methods and the vanilla (undefended model), we plot the dotted line originating from the vanilla results.

et al., 2017) with 100 epochs of standard training and 20 epochs of sparse tuning. For Texas100 and Purchase100, training is performed using MLPs as described in previous studies (Nasr et al., 2018; Jia et al., 2019), with 100 epochs of standard training and 20 epochs of sparse tuning.

**Hyperparameter Tuning**   In our approach to hyperparameter tuning, we align with the protocols established by previous work (Chen et al., 2022). In particular, we employ hyperparameter tuning focused on a single hyperparameter, $\alpha$ defined in Equation (4). Through a detailed grid search on a validation set, we adjust $\alpha$ to achieve an optimal balance. This process involves evaluating the privacy-utility implications at various levels of $\alpha$ and then selecting the value that aligns with our specific privacy/utility objectives, thereby enabling precise management of the model's privacy and utility. For the overall regularization strength $\lambda$ in Equation (3), we fix it to different values based on the dataset. Specifically, for CIFAR-10, the scale factor is 0.001; for CIFAR-100, it is 0.0001. For other datasets, we set it as 1e-05.

**Attack models**   In our study, we experiment with three classes of MIA: (1) Neural Network-based Attack (NN) (Shokri et al., 2017; Hu et al., 2022), which leverages the full logits prediction as input for attacking the neural network model. (2) Metric-based Attack, employing specific metric computations followed by a comparison with a preset threshold to ascertain the data record's membership. The metrics we chose for our experiments include Correctness (Correct), Loss (Yeom et al., 2018), Confidence, Entropy (Salem et al., 2019), and Modified-Entropy (M-entropy) (Song & Mittal, 2021). (3) Augmentation-based Attack (Choquette-Choo et al., 2021), utilizing prediction data derived through data augmentation techniques as inputs for a binary classifier model. In this category, we specifically implemented rotation and translation augmentations.

For the details of the attack, we assume the most powerful black-box adaptive attack scenario: the adversary has complete knowledge of our defense mechanism and selected hyperparameters. To implement this, we train shadow models with the same settings used for our target models.

**Defense baselines**   We compare PAST with eight defense methods: RelaxLoss (Chen et al., 2022), Mixup+MMD (Li et al., 2021), Adversarial Regularization (AdvReg) (Nasr et al., 2018),

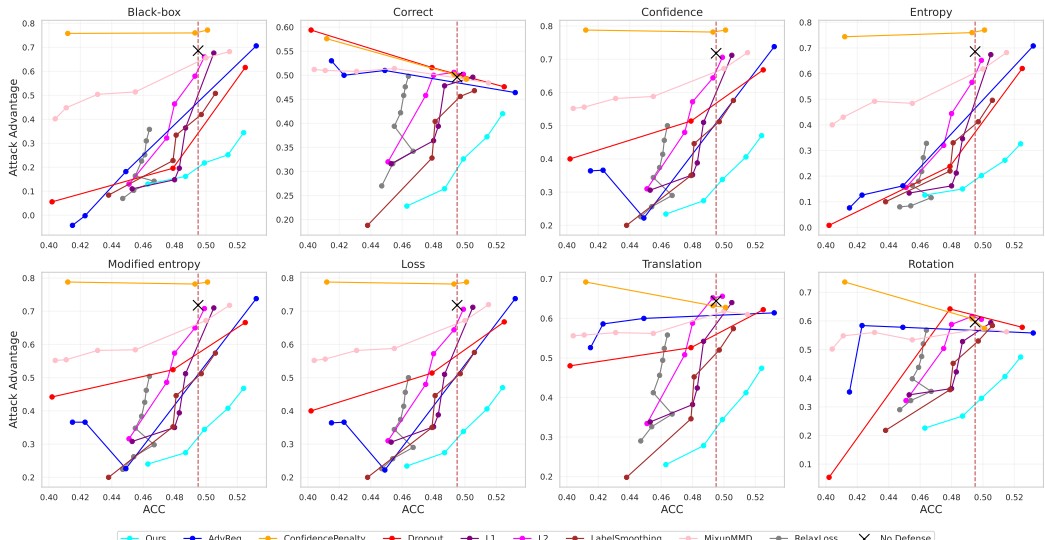

Figure 4: Comparisons of seven defense mechanisms on CIFAR-100 dataset utilizing Densenet121 architecture. Each subplot is allocated to a distinct attack method, wherein individual curves represent the performance of a defense mechanism under different hyperparameter settings. The horizontal axis represents the target models' test accuracy (the higher the better), and the vertical axis represents the corresponding attack advantage (defined in Definition 5, the lower the better). To underscore the disparity between the defense methods and the vanilla (undefended model), we plot the dotted line originating from the vanilla results.

Dropout Srivastava et al. (2014), Label Smoothing Guo et al. (2017), Confidence Penalty Pereyra et al. (2017), $\ell 1$ regularization and $\ell 2$ regularization (Shokri et al., 2017), .

**Evaluation metrics**  To comprehensively assess our method's impact on privacy and utility, we employ three evaluation metrics that encapsulate utility, privacy, and the balance between the two. Utility is gauged by the test accuracy of the target model. Privacy is measured through the attack advantage (Yeom et al., 2018):

$$
\begin{aligned}
Adv(\mathcal{A}) := &\Pr(\mathcal{A}(h_{\mathcal{S}}(\boldsymbol{x}), y) = 1 | m = 1) \\
&- \Pr(\mathcal{A}(h_{\mathcal{S}}(\boldsymbol{x}), y) = 1 | m = 0) \\
= &\, 2 \Pr(\mathcal{A}(h_{\mathcal{S}}(\boldsymbol{x}), y) = m) - 1
\end{aligned}
\tag{5}
$$

where the notations are defined in Section 2.1. To assess the trade-off between utility and privacy, we utilize the $P_1$ score (Paul et al., 2021), which is defined as:

$$
P_1 = 2 \times \frac{\text{Acc} \times (1 - \text{Adv})}{\text{Acc} + (1 - \text{Adv})}
\tag{6}
$$

where Acc denotes test accuracy and Adv denotes attack advantage on the target model.

### 4.2 RESULTS

**Can PAST improve privacy-utility trade-off ?**  In Figure 3 and Figure 4, we plot privacy-utility curves to show the privacy-utility trade-off. The horizontal axis represents the performance of the target model, and the vertical axis represents the attack advantage defined in Equation (5). A salient observation is that our method drastically improves the privacy-utility trade-off. In particular, for these points that perform better than vanilla for utility (the area to the right of the dotted line), the privacy-utility curves of our methods are always below those of others. This means we can always obtain the highest privacy for any utility requirement higher than the undefended model. For example, on the CIFAR10, we focus on the hyperparameter $\alpha$ corresponding to the model with the lowest attack advantage with the constrain condition that test accuracy is better than vanilla, then our method with adaptive regularization can decrease the attack advantage of loss-metric-based from 14.8% to 5.2% compared with MixupMMD (the most powerful defense under our condition above).

Table 1: P1 score (Equation (6)) evaluated on target models trained on different datasets. The bold indicates the best results. Here, "w/o" denotes undefended models.

| Datasets | Texas | Purchase | CIFAR-10 | CIFAR-100 | ImageNet |
|---|---|---|---|---|---|
| PAST | **0.572** | **0.812** | **0.784** | **0.575** | **0.438** |
| w/o | 0.557 | 0.792 | 0.638 | 0.360 | 0.350 |

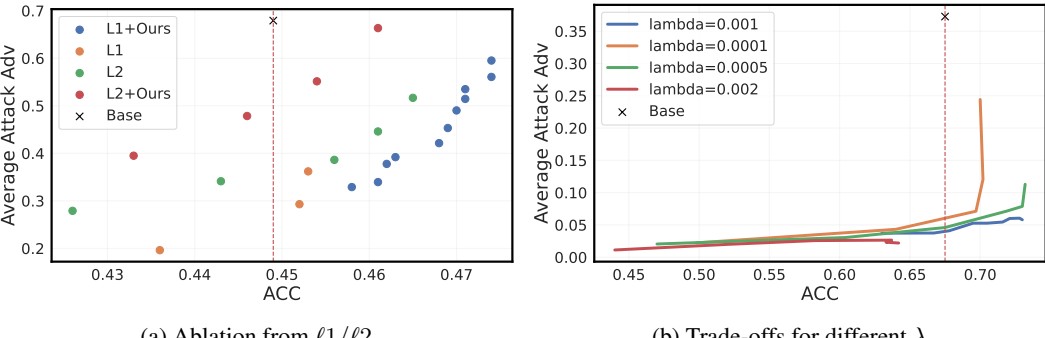

(a) Ablation from $\ell1/\ell2$        (b) Trade-offs for different $\lambda$

Figure 5: (a) Utility-privacy trade-offs for fixed/ours adaptive weights and $\ell1/\ell2$ regularizations. Dots in each color represent the performance of a tuning mechanism under different hyperparameter settings. The horizontal axis represents the test accuracy (the higher the better), and the vertical axis represents the average attack advantage (defined in Definition 5, the lower the better. Results w/o average are in Appendix A) across various attack methods. PAST (L1+Ours) outperformed others; (b) Utility-privacy trade-offs (by tuning $\alpha$) for different $\lambda$. The x-axis and y-axis are the same as (a). Within a certain range ($\lambda = 0.0005, 0.001$ here), the trade-off curve remains stable.

**Is PAST effective with different datasets?** To ascertain the efficacy of our proposed method across heterogeneous data, we have executed a series of experiments on a diverse array of datasets, encompassing tabular and image datasets. For results shown in Table 1, we have set the adjustment $\alpha$ of PAST to a constant value, specifically $\alpha = 2.5$. To assess the privacy-utility balanced performance, we use the highest attack advantage of all attack methods to calculate the P1 score From the results, we observe that both of our methods yield a consistent improvement in the P1 score.

**How does $\alpha$ affect utility and privacy?** In Figure 6a, we conduct an ablation study to examine the impact of the coefficient $\alpha$ in our method on both utility and privacy (and the effect on loss gap is reported and analyzed in Appendix C). The analysis is based on CIFAR-100. As is shown in Figure 6a, our findings are in alignment with the insights provided in Section 3.2. As the $\alpha$ decreases, the effect of the loss gap becomes less significant, leading to a gradual decrease in adaptation strength. On the other hand, a smaller $\alpha$ value brings our loss function closer to the conventional regularization, thereby increasing the privacy risk. Conversely, A larger $\alpha$ leads to stronger regularization on sensitive parameters, culminating in underfitting, which consequently diminishes accuracy.

**What's the difference between PAST and $\ell1/\ell2$ regularization?** We compare our method with $\ell1/\ell2$ regularization on CIFAR-100 (by fixing the adaptive weight $\gamma_i$ in Equation (3) to a constant ), and present the results in Figure 5a. Specifically, we used four combinations during tuning: $\ell1$ regularization (L1), $\ell1$ regularization + adaptive weight (L1+Ours), $\ell2$ regularization (L2), and $\ell2$ regularization + adaptive weight (L2+Ours). For the $\ell1/\ell2$ regularization, we adjust the regularization weight $\lambda$ to achieve the desired utility-privacy trade-off. Dots of each color in Figure 5a represent the performance of a tuning mechanism under different hyperparameter settings. As observed, PAST (L1+Ours) outperformed the others. This demonstrates the importance of incorporating adaptive regularization weights in achieving robust defense against MIAs.

**How does $\lambda$ affect PAST?** $\lambda$ in Equation (4) represents a base level of regularization applied to all weights, similar to the influence in $\ell1$ regularization. We fixed $\lambda$ at different values ($\{0.0001,$

Table 2: P1 score (defined in Equation (6)) evaluated on target models tuned on defended models. The bold indicates the best results. Here, "w/o" denotes the original defended model by other methods. PAST consistently achieves higher P1 scores compared to the original defended methods

| Pretrain Defense | AdvReg | CCL | LabelSmoothing | MixupMMD | RelaxLoss |
|---|---|---|---|---|---|
| PAST | **0.784** | **0.806** | **0.808** | **0.825** | **0.808** |
| w/o PAST | 0.720 | 0.657 | 0.674 | 0.755 | 0.744 |

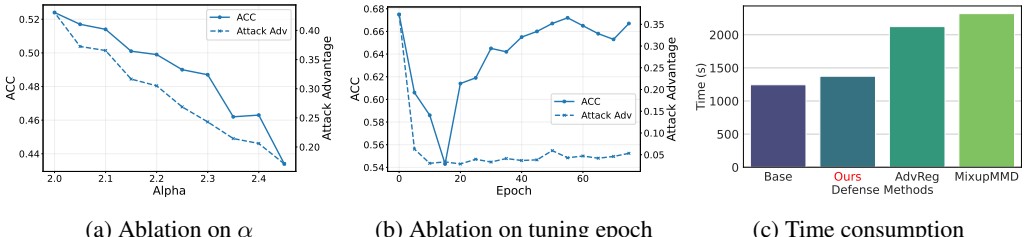

| (a) Ablation on $\alpha$ | (b) Ablation on tuning epoch | (c) Time consumption |
|---|---|---|

Figure 6: (a) Effect of $\alpha$ on utility (test accuracy) and privacy (average attack advantage). Both accuracy (ACC) and attack advantage decrease as alpha increases, in alignment with the insights provided in 3.2; (b) Effect of tuning epochs on utility and privacy. The defense effectiveness stabilizes at an optimal level after 20 epochs, and the classification accuracy gradually improves, peaking at around 50 epochs; (c) Time consumption of PAST and other methods. Each bar stands for a method, Ours are comparable to standard training(Base)

0.0005, 0.001, 0.002}) on CIFAR-10 and adjusted alpha to plot a utility-privacy curve for each $\lambda$ (In Figure 5b). Within a certain range ($\lambda = 0.0005, 0.001$ here), the trade-off curve remains stable; however, when $\lambda$ is too large, the trade-off cannot achieve high utility (e.g., $\lambda = 0.002$ here). On the other hand, when $\lambda$ is too small, the overall regularization strength is too weak, resulting in an effect closer to the base, which leads to higher privacy leakage (e.g., $\lambda = 0.0001$ here)

**How does the number of tuning epochs affect PAST?** We evaluated the impact of different sparse training epochs on the utility-privacy trade-off (the effect on loss gap is reported and analyzed in Appendix C). Specifically, we conducted experiments on CIFAR-10, varying the number of epochs across $\{5, 10, \cdots, 75\}$. The curves of test accuracy and attack advantage over epochs are plotted in Figure 6b, with the dotted line representing attack advantage and the solid line representing test accuracy. As the number of epochs increases, the advantage stabilizes at an optimal level after 20 epochs, and the classification accuracy gradually improves, peaking at around 50 epochs. Overall, it shows that few epochs are sufficient to achieve reasonable performance, and more epochs lead to more stable outcomes.

**Combined with other defenses** As mentioned in Section 3.2, our method is applied during the fine-tuning phase, as the loss gap can more accurately reflect member information in a roughly converged model. Due to this characteristic, our approach is independent of other defense methods applied during the pre-training phase and can be used on top of existing defenses. We conducted experiments with various defense methods on CIFAR-10 using ResNet18 (50 epochs sparse training with $\alpha = 1.5$). As shown in Table 2, after PAST, the model consistently achieves higher P1 scores compared to the original defense methods.

**Time consumption** The time consumption of PAST is comparable to standard training, since it introduces no additional processes (it only requires an extra gradient backpropagation during each tuning epoch to obtain the gradients for non-members). We report the time consumption of various defense methods in Figure 6c (time was recorded for DenseNet121 on CIFAR-100 using a single RTX 4090 GPU.), where our method takes 1374 seconds, and the standard training (base) takes 1245 seconds. Our approach increases the time consumption of standard training by only 10.4%.

## 5 RELATED WORK

**Overparameterization in generalization and privacy**  Overparameterization, where models have significantly more parameters than training examples, has been shown to have a complex relationship with generalization and privacy. While traditional theories suggest that overparameterization increases overfitting and generalization error, recent research reveals that it can sometimes reduce error under certain conditions, such as in high-dimensional ridgeless least squares problems (Belkin et al., 2020). This phenomenon, known as "double descent", suggests that beyond a critical point, increasing model complexity may lead to better generalization (Belkin et al., 2019; Dar et al., 2021; Hastie et al., 2022). However, from a privacy perspective, overparameterization has been empirically proven to increase vulnerability to membership inference attacks (MIAs) (Leemann et al., 2023; Dionysiou & Athanasopoulos, 2023). Large language models, in particular, are susceptible to these attacks, with attackers able to extract sensitive training data (Carlini et al., 2021; Mireshghallah et al., 2022). Theoretical evidence also indicates that there is a clear parameter-privacy trade-off, where an increase in the number of parameters amplifies the privacy risks by enhancing model memorization (Yeom et al., 2018; Tan et al., 2022b). Consequently, while overparameterization can sometimes improve generalization, its impact on privacy remains a significant concern, especially in the context of MIAs.

**Overparameterization in MIA defenses**  To mitigate the privacy risks associated with overparameterization, several defense mechanisms have been proposed. One effective approach is network pruning, where unnecessary parameters are removed to reduce model complexity. Research shows that pruning not only preserves utility but also significantly reduces the risk of privacy leakage, in scenarios including MIA (Huang et al., 2020; Wang et al., 2021) and Unlearning (Hooker et al., 2019; Wang et al., 2022; Ye et al., 2022b; Liu et al., 2024a). Additionally, techniques combining pruning with federated unlearning have demonstrated effectiveness in protecting privacy by selectively forgetting specific data during the training process (Wang et al., 2022). Regularization methods, such as $\ell 2$ regularization (Kaya et al., 2020), sparsification (Bagmar et al., 2021) and dropout (Galinkin, 2021), also play a critical role in defending against MIAs by discouraging the model from overfitting to training data. Interestingly, while overparameterization generally increases privacy risks, when paired with appropriate regularization, it can maintain both utility and privacy (Tan et al., 2023). Furthermore, studies indicate that initialization strategies and ensemble methods can further alleviate privacy risks on over-parameterized model (Rezaei et al., 2021; Ye et al., 2024). These techniques illustrate that even in overparameterized models, privacy risks can be mitigated through careful design, preserving the balance between utility and privacy.

## 6 CONCLUSION

In this paper, we introduce Privacy-aware Sparsity Tuning (PAST), a novel approach to mitigating membership inference attacks (MIAs) by adaptively regularizing model parameters based on the loss gap between member and non-member data. By promoting sparsity in parameters with large privacy sensitivity, the model shrinks the loss gap between members and non-members, leading to strong resistance to privacy attacks. Extensive experiments demonstrate that PAST effectively balances privacy and utility, providing state-of-the-art performance in the privacy-utility trade-off. This method is straightforward to implement with existing deep learning frameworks and requires minimal modifications to the training scheme. We hope that our insights into Privacy-aware regularization inspire further research to explore parameter regularization techniques for enhancing privacy in machine learning models.

**Limitations**  In this work, we focus on the popular black-box setting, where attackers can access the model outputs. So, the effectiveness of our method in defending against other types of MIAs (such as label-only attacks, white-box attacks) remains unexplored. Moreover, while our method can improve the MIA defense with high predictive performance, our method cannot fully break the trade-off between utility and MIA defense, which might be a potential direction for future work.

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

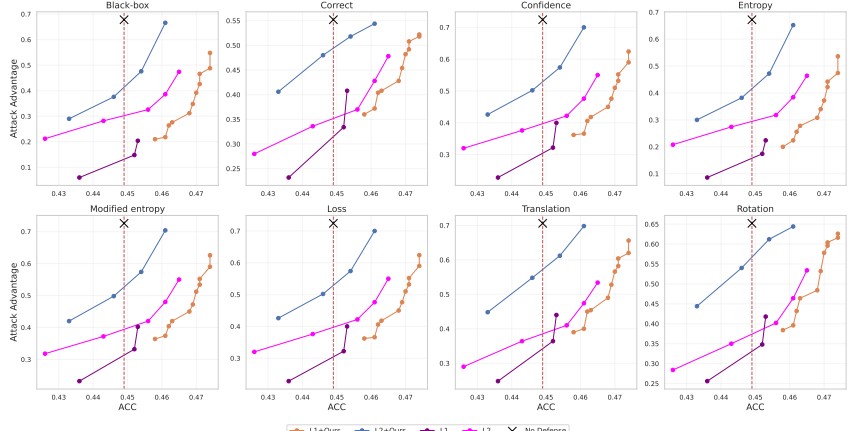

Figure 7: Utility-privacy trade-offs of different epochs on CIFAR-10. Each subplot is allocated to a distinct attack method, wherein individual curves represent the performance of a defense mechanism under different hyperparameter settings. The horizontal axis represents the target models' test accuracy (the higher the better), and the vertical axis represents the corresponding attack advantage (defined in Definition 5, the lower the better). To underscore the disparity between the defense methods and the vanilla (undefended model), we plot the dotted line originating from the vanilla results.

## A    FULL ABLATION RESULT FOR PAST

Here in Figure 7, we present the detailed results of ablation study (fixed/ours adaptive weights and $\ell 1/\ell 2$ regularizations). under different attack methods on CIFAR-100 (in the main text, only the average performance across various attack methods is shown due to layout constraints). It can be observed that the performance varies under different attack methods, but the overall utility-privacy trade-off of PAST evidently surpasses others.

## B    DIFFERENCE FROM OPTIMIZING TO LOSS GAP

In this section, we compare the differences between PAST and directly optimizing the loss gap. We use the loss gap between members and non-members directly as a regularization term and add it to the loss function. The utility-privacy trade-off curves are shown in Figure 8, where our method (Ours) clearly outperforms the direct optimization of the loss gap (Loss-Gap). Additionally, we point out that directly optimizing the loss gap leads to privacy leakage in the inference set (as shown by LossGap (infer) in the figure), whereas our method does not (as shown by Ours (infer) in the figure).

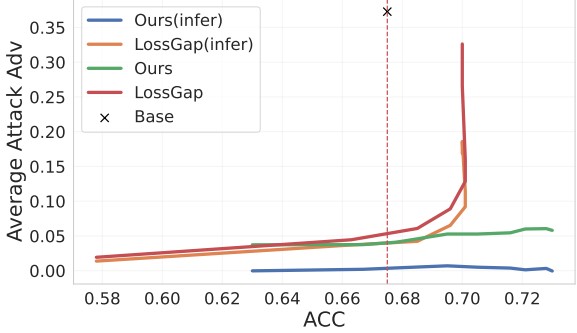

Figure 8: The utility-privacy trade-offs of Ours and LossGap (directly optimizing member-nonmember loss gap), "infer" refer to the privacy leakage of inference set

## C    LOSS GAP FOR DIFFERENT EPOCHS/ALPHAS

In the main text, the ablation on tuning the epoch and alpha only reports the changes in utility (test accuracy) and privacy (attack advantage). Here, we supplement with the changes in the privacy proxy loss gap as alpha (in Figure 9a) and epoch (in Figure 9b) varies, showing that the results here are consistent with the insights provided in Section 3.2.

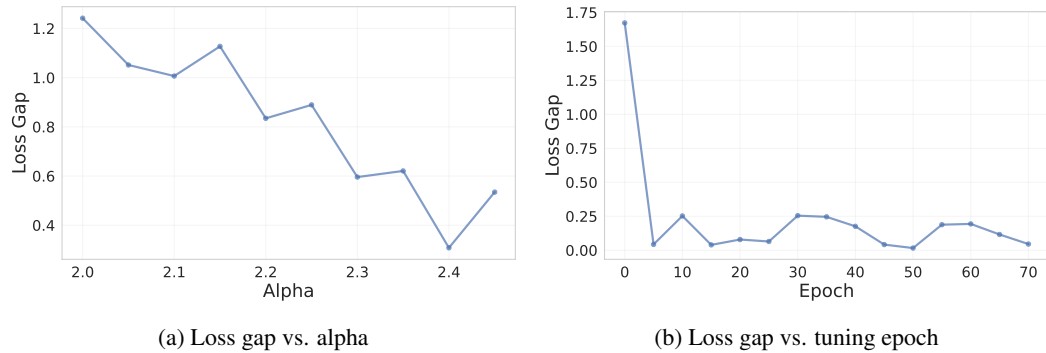

(a) Loss gap vs. alpha

(b) Loss gap vs. tuning epoch

Figure 9: (a) The member-nonmember loss gap varies with different alpha values. As alpha increases, the loss gap continuously decreases, validating the effect of PAST in shrinking the loss gap. (b) The loss gap varies with different epochs. As the number of epochs increases, the loss gap decreases rapidly, reaching nearly its minimum after just 5 epochs. This demonstrates that PAST's effect in shrinking the loss gap can be achieved with only a few epochs.

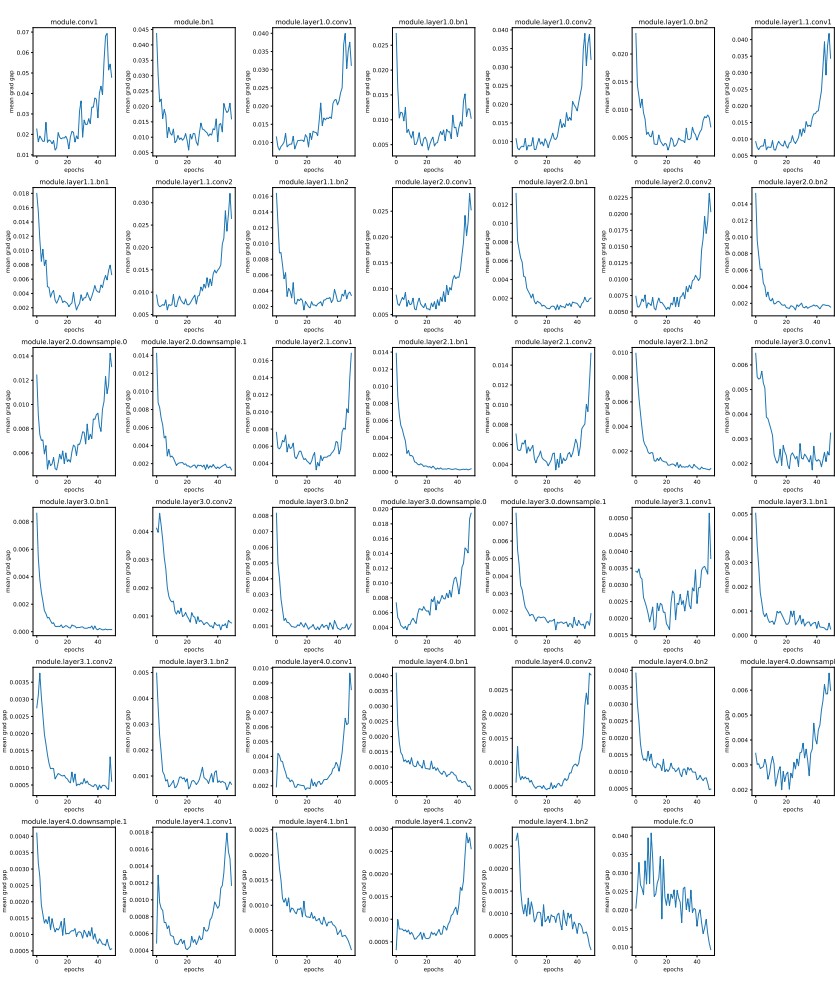

Figure 10: Variation of privacy sensitivity for each module in PAST

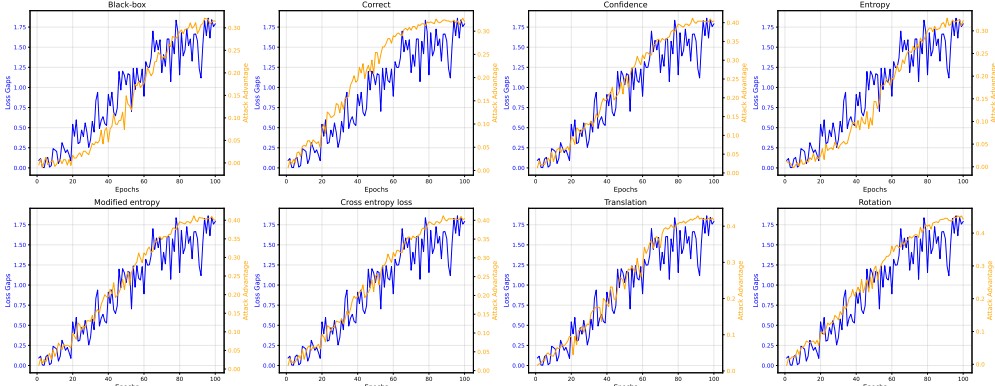

Figure 11: The variation of the loss gap and the attack advantage across various attacks during standard training. During standard training, the loss gap monotonically increases in sync with the attack advantage across various attacks. This indicates that the loss gap, as a privacy proxy, can effectively capture the effects of various attacks.

## D    FOR WHICH LAYERS AND MODULES ARE MORE EFFECTIVE?

As an example, we illustrate the average grad-gap dynamics during the PAST process across different modules of the ResNet18 model in Figure 10. It can be observed that the deeper layers are more effective than the earlier ones, and batch normalization (BN) and linear layers contribute more significantly than convolutional layers. Notably, the loss gap of all convolutional layers in the third and fourth blocks almost stabilizes at zero.

## E    RATIONALE AND GENERALIZABILITY OF THE LOSS GAP AS A PRIVACY PROXY.

To clarify the rationale for using the loss gap as a proxy risk for privacy, we theoretically characterize that the loss gap is positively correlated with the attack advantage.

**Proposition E.1.** *Let $\epsilon$ be a random variable denoting loss, such that $\epsilon \sim N(\mu_S, \sigma_S^2)$ when $m = 1$ and $\epsilon \sim N(\mu_D, \sigma_D^2)$ when $m = 0$. Then the loss gap $(\mu_D - \mu_S)$ is positively correlated with the attack advantage, defined in Equation (5).*

*Proof.* The membership advantage of $\mathcal{A}_{\text{loss}}$ is (as defined in Equation (5)):

$$Adv = \Pr(\mathcal{A} = 1 | m = 1) - \Pr(\mathcal{A} = 1 | m = 0) \tag{7}$$

$$= \Pr(\epsilon \leqslant \tau | m = 1) - \Pr(\epsilon \leqslant \tau | m = 0) \tag{8}$$

$$= \Phi(\frac{\tau - \mu_S}{\sigma_S}) - \Phi(\frac{\tau - \mu_D}{\sigma_D}) \tag{9}$$

where $\Phi(\cdot)$ is the cumulative distribution function of standard normal distribution. Note that $\Pr(\mathcal{A} = 1 | m = 0)$ is false positive rates of the adversary, which is expected to be controlled at a small value Leemann et al. (2023); Tan et al. (2022b). Assume $\tau$ is chosen such that $\Phi(\frac{\tau - \mu_D}{\sigma_D}) = \alpha$, then we have:

$$Adv = \Phi\{\frac{\Phi^{-1}(\alpha)\sigma_D + \mu_D - \mu_S}{\sigma_S}\} - \alpha \tag{10}$$

Since $\frac{\partial(Adv)}{\partial(\mu_D - \mu_S)} = \frac{1}{\sigma_S}\phi\{\frac{\Phi^{-1}(\alpha)\sigma_D + \mu_D - \mu_S}{\sigma_S}\} > 0$, this implies that the loss gap $(\mu_D - \mu_S)$ is positively correlated with the attack advantage $Adv$.  $\square$

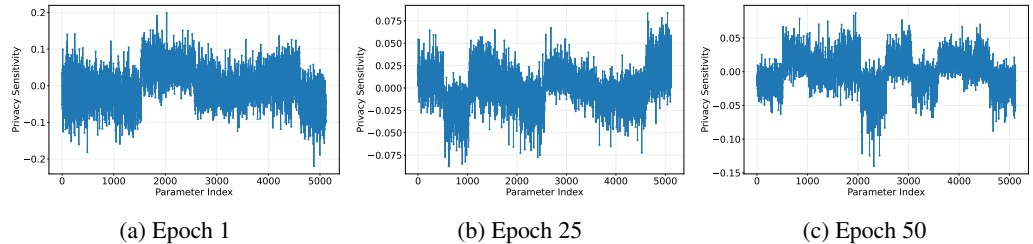

(a) Epoch 1          (b) Epoch 25          (c) Epoch 50

Figure 12: The change of privacy sensitivity distribution during PAST. We fixed the x-axis as parameter index and the y-axis as privacy sensitivity, reporting results for epochs 1, 25, and 50. It can be observed that privacy sensitivity indeed migrates, while the overall privacy sensitivity decreases over time.

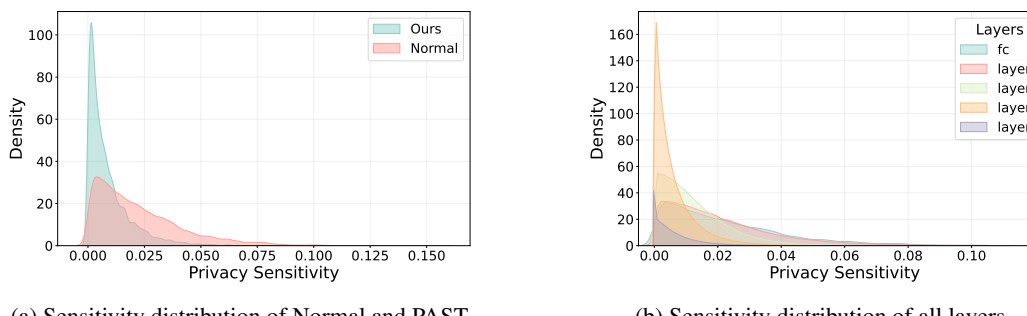

(a) Sensitivity distribution of Normal and PAST      (b) Sensitivity distribution of all layers

Figure 13: (a) The sensitivity distribution of FC parameters for standard training (Normal) and PAST. PAST has fewer privacy-sensitive parameters, indicating that it remains effective even if privacy migration occurs, as the overall sensitivity is reduced. (b) The sensitivity distribution of parameters across all layers (layer 1-4 and the fc layer). Within each layer, the finding that "only a small fraction of parameters substantially impacts privacy risk" remains significant. This indicates that the finding is not merely due to parameters closer to the output layer naturally having more influence on gradients and results.

In terms of generalizability of using the loss gap as a proxy for privacy risk, we empirically demonstrate the effectiveness of the loss gap as a privacy proxy by showing its relationship with the attack advantage across various attack methods. Specifically, in Figures 1a and 11, we observe that during standard training, the loss gap monotonically increases in sync with the attack advantage, suggesting that the loss gap can capture different aspects of the model's output and behavior.

# F  ADDITIONAL RESULTS

## F.1  DOES PRIVACY MIGRATE AMONG PARAMETERS?

We analyzed the changes in privacy sensitivity during the PAST training process and confirmed the existence of privacy migration. Specifically, we plotted the distribution of privacy sensitivity (y-axis) against parameter index (x-axis) at the 1st, 25th, and 50th epochs during training on the fully connected (fc) layer of ResNet18 trained on CIFAR-10. The results, shown in Figure 12, confirm that privacy migrates from heavily regularized parameters to others during PAST.

The migration of privacy sensitivity does not affect the effectiveness of our method. Our proposed regularization focuses on parameters deemed "important" in each epoch, and these parameters can change dynamically. As shown in Figure 13a, we compare the privacy sensitivity distributions of models trained with PAST versus standard training. The results indicate a significant reduction in the overall privacy sensitivity (with the mean reduced from 0.0223 to 0.0088), demonstrating the robustness of our method in mitigating privacy risks.

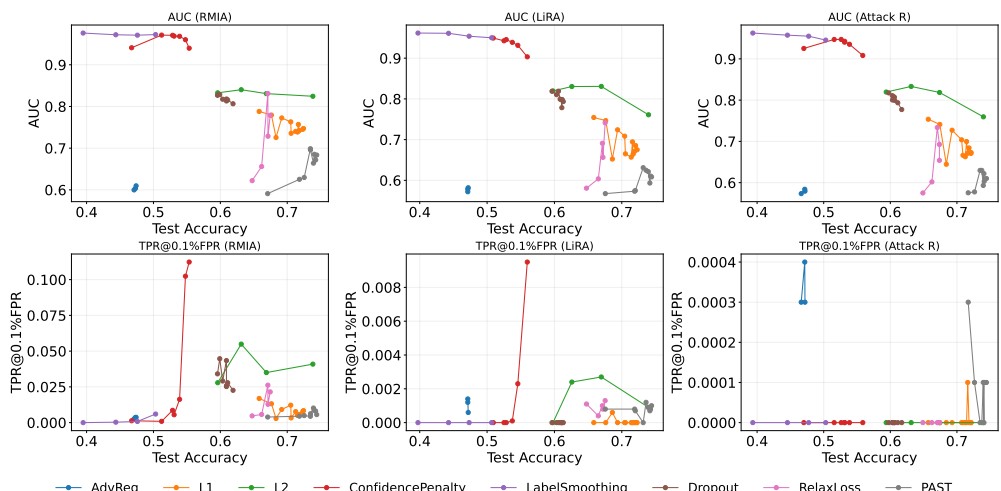

Figure 14: AUC vs. Test Accuracy curves and TPR@0.1%FPR vs. Test Accuracy curves for RMIA (Zarifzadeh et al., 2024), LiRA (Carlini et al., 2022) and Attack R (Ye et al., 2022a). The x-axis represents Test Acc (higher is better), and the y-axis represents attack effectiveness (lower is better). It can be observed that PAST achieves a trade-off positioned in the bottom right, demonstrating strong performance.

## F.2    PRIVACY SENSITIVITY WITHIN EACH LAYER

In this section, we provide empirical evidence to show that the phenomenon—where only a small fraction of parameters substantially impacts privacy risk—exists within each layer of the neural network, rather than being solely attributed to differences between layers. Using a ResNet18 model trained on CIFAR-10 as an example, we demonstrate the privacy sensitivity distribution of parameters within each layer, as shown in Figure 13b. The observation that "only a small fraction of parameters substantially impacts privacy risk" holds true within each layer, indicating that this finding is not merely due to natural differences in gradients across layers.

## F.3    ADDITIONAL RESULTS OF ATTACK METHODS AND EVALUATION METRICS

We have added results for new MIA methods: RMIA (Zarifzadeh et al., 2024), LiRA (Carlini et al., 2022) and Attack R (Ye et al., 2022a). Here, following the setting in RMIA, we conducted experiments on ResNet18 trained on CIFAR-10, using AUC and TPR@0.1%FPR as metrics. By tuning the hyperparameters of each defense method, we plot utility-privacy trade-off curves in Figure 14, where the horizontal axis represents the target models' test accuracy (the higher the better), and the vertical axis represents the corresponding metric (AUC or TPR@0.1%FPR, the lower the better). It can be observed that PAST demonstrates strong performance under both metrics and outperforms other defenses.

