# OpenReview forum: "Defending Membership Inference Attacks via Privacy-aware Sparsity Tuning"
_ICLR.cc/2025/Conference — Submitted to ICLR 2025_

### Official Review · Reviewer_5igj · 2024-11-02

**Soundness:** 2
**Presentation:** 2
**Contribution:** 2
**Rating:** 5
**Confidence:** 4

**Summary:**

The paper presents Privacy-Aware Sparsity Tuning (PAST), an approach to defending against membership inference attacks (MIAs) by introducing an adaptive regularization method that selectively applies sparsity based on privacy sensitivity of parameters. The proposed method modifies ℓ1 regularization, assigning higher penalties to parameters with greater privacy sensitivity. The authors conduct experiments across datasets (Texas100, Purchase100, CIFAR-10/100, and ImageNet) to validate the efficacy of PAST, reporting improvements in the privacy-utility trade-off over baseline defenses.

**Strengths:**

Clarity: The paper is easy to read.

Relevance: The paper aims to address an important topic in machine learning privacy.

**Weaknesses:**

1.Lack of Comparison with Recent Baselines: The current version does not compare its approach with recent baselines from the past three years, which is crucial for an ICLR 2025 submission. The paper only references MIA attacks from six years ago and defenses from three years ago, missing key recent works that are important for establishing state-of-the-art (SOTA) context. Key references to consider include:

MIA Attacks:

(a)"Low-Cost High-Power Membership Inference Attacks" (ICML 2024)

(b)"Scalable Membership Inference Attacks via Quantile Regression" (NeurIPS 2023)

(c)"Membership Inference Attacks from First Principles" (S&P 2022)

(d)"Enhanced Membership Inference Attacks against Machine Learning Models" (CCS 2022)

MIA Defenses:

(a) "MIST: Defending Against Membership Inference Attacks Through Membership-Invariant Subspace Training" (USENIX 2024)

(b) "SELENA: Mitigating Membership Inference Attacks by Self-Distillation Through a Novel Ensemble Architecture" (USENIX 2022)

These references represent a minimum set of recent works that should be discussed and included as baselines for a credible comparison.

2.Method Simplicity and Lack of Theoretical Rigor: The proposed method relies on a heuristic reweighting approach that lacks theoretical guarantees, which is a notable limitation for top conference standards. The reweighting method in Privacy-Aware Sparsity Tuning (PAST) is too simplistic and does not reach the research depth, completeness, and robustness generally expected of accepted MIA papers in top conferences. The authors are encouraged to enhance the theoretical foundation and complexity of their approach for future submissions.

3.Weak Motivation and Contribution: The paper's motivation, as shown in Figure 1(b) ("Only a small fraction of parameters substantially impacts privacy risk"), is already a well-known concept in machine learning. Given that parameters closer to the output layer naturally have more influence on gradients and results, this finding is not novel. The use of regularization as an MIA defense is also limited, as regularization is not a commonly adopted SOTA defense method in MIA. The paper does not address whether the proposed reweighting improvement would benefit more commonly used defenses, making its contribution less impactful given that it builds on a sub-optimal defense method.

**Questions:**

See weaknesses for details. I am giving a score of 5 (marginally below the acceptance threshold). A lower score of 3 would feel overly harsh, as the paper shows some promise. However, this version still falls short of the standards expected at a top conference like ICLR. If the authors remain enthusiastic about this work and believe in the effectiveness of their approach, I encourage them to address the above feedback and consider resubmitting to a future ML or security-focused conference. The current version, however, is far from reaching the level required for acceptance at a top-tier conference.

---

> ### Author Response · Authors · 2024-11-23
> **Response to Reviewer 5igj**
>
> Thank you for the recognition and valuable comments. Please find our response below.
>
> **1. Lack of comparison with recent baselines [W1]**
>
> Thank you for the suggestion. Please find the detailed analysis in the General Response.
>
> **2. Method simplicity and lack of theoretical rigor [W2]**
>
> Thank you for the suggestion. We add a theoretical motivation for the choice of the loss gap as a privacy proxy in Appendix E. In addition, we politely disagree that the simple design of our method is a drawback. The contribution of a machine learning paper depends on the **insight** provided in the paper, instead of the complexity of the method. Generally, a **simple and effective** method is preferred (as appreciated by reviewer DxZ7), as it can easily validate the insight and obtain higher impacts in the community.
>
> In this work, we provide the unique insight: only a small fraction of parameters substantially impacts the privacy risk. Thus, we design a simple and effective method to promote sparsify on those "important" parameters. We believe the contribution is sufficient for a top ML conference.
>
> **3. Concerns of novelty and contribution [W3]**
>
> We politely disagree that our conclusion is already a well-known concept in machine learning. To be professional, we would **provide a citation to existing works** if we claim something is not novel as a reviewer. We provide the response for each concern raised by the reviewer.
> * **The influence of layers.** In Appendix F.2, we provide empirical results to show that this pheonomenon exists in each layer of neural network: Among the parameters from the same layer, only a small fraction of them substantially impacts privacy risk. This demonstrates that our finding is not due to the difference between layers. Besides, we employ a normalization in Eq.(4) to mitigate the discrepancy among different parameter modules in our method.
> * **SOTA attacks.** In Appendix F.3, we add new results to show that our method can perform the best in defending the recent SOTA attacks -- **RMIA** [1] (ICML 2024), **LiRA** [2] (S&P 2022) and **Attack R** [3] (CCS 2022). This shows the role of regularization as an effective method in defending MIAs (at least from now).
> * **Benefits to common defense methods.** In  table 2, we demonstrated in Table 2 that PAST can enhance other common defense methods.
>
> In summary, the insight shown in this paper is novel and can inspire more specific-designed methods in the future. Our method performs the best in defending the recent SOTA attacks and can benefits common defense methods. With the new insight and strong method, our work makes sufficient contribution to the community.
>
> [1] Zarifzadeh, et al. Low-Cost High-Power Membership Inference Attacks. ICML 2024.
>
> [2] Carlini, N., et al. Membership inference attacks from first principles. S&P 2022.
>
> [3] Ye, J., et al. Enhanced membership inference attacks against machine learning models. CCS 2022.

---

### Official Review · Reviewer_8dBT · 2024-11-03

**Soundness:** 2
**Presentation:** 4
**Contribution:** 2
**Rating:** 3
**Confidence:** 4

**Summary:**

This paper proposes an approach called PAST. Instead of using uniform penalties, the PAST identifies and raises sparsity in parameters, which significantly contributes to privacy leakages. The paper uses the loss gap as the functional proxy of the privacy loss captured by the attack advantage. The weight for each parameter is set adaptively based on its privacy sensitivity defined as the gradient of the loss gap with respect to the parameter.

**Strengths:**

This paper acknowledges the different impacts of parameters on privacy leakage and proposes an adaptive approach to promote privacy-aware weight specifications for each parameter. This is an important consideration for privacy protection. In addition, the paper involves extensive numerical experiments to show the superiority of the proposed PAST.

**Weaknesses:**

There are flaws and issues in the motivation of considering the loss gap, the consistency between the attacker advantage and the loss gap as the functional proxy of privacy risk, the experimental setup, and the characterizations related to the privacy-utility tradeoff.

I would be open to increasing the score if the authors can adequately address my concerns during the rebuttal.

Comment 1:

The rationale for using the loss gap as a proxy risk for privacy is unclear. Specifically, the referenced paper (Sablayrolles et al., 2019) considers a specific case in which the posterior in Eq. (1) is modeled as a Boltzmann distribution, creating a direct link between the loss function and the posterior distribution. Generally, the posterior distribution is crucial in characterizing privacy risk. The work by Sablayrolles et al. (2019) assumes a direct connection between the loss function and privacy risk, with all results depending on this assumption, which does not extend to general loss functions. Therefore, using the statement, “It has been shown that the loss function can determine the optimal attacks in membership inference (Sablayrolles et al., 2019),” as justification for adopting loss gaps as a proxy for privacy loss is questionable and insufficient. In addition, the paper does not theoretically or analytically characterize the rationale for using the loss gaps as a proxy of the privacy risk in terms of attacker advantage.

In experiments, three classes of MIA models are considered. However, the use of the loss gap as a proxy for privacy risk does not fully coincide with the privacy risk induced by these MIA models. In particular, the loss gap captures the average loss distinguishability between the member and the non-member. However, different MIA models including the three considered use a range of observations, including logits, confidence, entropy, correctness, and augmented predictions. These observations (of the attacker) capture different aspects of the mechanism's/model's output and behavior beyond just the loss values. As a result, the privacy risk represented by these attacks may involve patterns in these observations that a simple loss gap does not fully capture. For example, the augmentation-based attack depends on how the model responds to augmented inputs, which can reveal membership through consistency in the model's behavior across transformations. Loss gap does not capture this aspect of the model's sensitivity to augmented variations, so it might underestimate privacy risk in such scenarios.

Thus, it is in general possible for two different weight schemes with similar loss gaps to exhibit notably different attacker advantages.

Comment 2:

My concern regarding the feasibility of using loss gap as a proxy of privacy risk leads to the concern of adaptively setting the parameter weights based on the privacy sensitivity (i.e., the gradient of the loss gap w.r.t. the parameter). That is, the gradient might not precisely reflect the privacy sensitivity. It is suggested to provide analytical and theoretical characterizations for the feasibility of using loss gap as a proxy of privacy risk, including the possible assumptions.


Comment 3:

The comparison between the PAST and the uniform-penality-based $\ell_1$ and $\ell_2$ baselines (or uniform baselines) needs more details. The paper clearly describes the difference between PAST and uniform baselines in terms of whether the weights of the parameters are uniform or privacy-aware. However, the impact of the magnitude of the weights on privacy risk is ignored (correct me if I am missing anything).

Q1: In Figure 3 and Figure 4, what are the values of lambdas chosen for $\ell_1$ and $\ell_2$ baselines?

Q2: Is it possible that the performance differences between PAST and the $\ell_1$ and $\ell_2$ baselines shown in Figure 3 and Figure 4 are due to the combination of the magnitude of the weights and setting of uniform or privacy-aware penalties, rather than due to the setting of uniform penalties or privacy-aware alone?

It is intuitive that properly weighting the more-privacy-sensitive parameters more than the less-privacy-sensitive parameters might be enough to guarantee a certain degree of privacy. However, the comparison of PAST and the $\ell_1$ and $\ell_2$ baselines in terms of the privacy-utility/accuracy tradeoff without considering the magnitude of the weights could lead to unfair comparisons.

Comment 4:

Overparameterization is not the same as non-properly weighting a fixed number of parameters.  The proposed PAST applies adaptive regularization by adjusting the penalty weights of parameters based on their privacy sensitivity, which does not necessarily address the issue of overparameterization.

Section 5 RELATED WORK discusses the relationship between the number of parameters (or model complexity) and privacy risk, particularly in the context of overparameterization. There are works that confirm the tradeoff between the number of parameters and privacy risk. However, there is limited discussion of how the specific values or magnitudes of those parameter weights might impact privacy. Adjusting the weights of parameters for fixed parameters could influence the model's tendency to memorize or generalize, which potentially affects privacy risk. However, the related work section does not discuss studies that study this trade-off between parameter weight and privacy directly. In addition, the paper is suggested to characterize this weight-privacy trade-off.

**Questions:**

See Weakness.

---

> ### Author Response · Authors · 2024-11-23
> **Response to Reviewer 8dBT**
>
> We appreciate the reviewer for the insightful and detailed comments. Please find our response below:
>
> **1. The rationale for using the loss gap as a proxy risk [W1, W2]**
>
> Thank you for raising the concern. To clarify the rationale for using the loss gap as a proxy risk for privacy, we theoretically characterize that the loss gap is positively correlated with the attack advantage. The proof is provided in Appendix E.
>
> In terms of generalizability of using the loss gap as a proxy risk, we argue that it is impractical to formally identify a perfect proxy for all potential attack methods. In this paper, we empirically validate the generalizability of our method, by showing the effective defend against various attack methods. Besides, we show, in Figure 1(a) and Figure 11, that both the loss gap and the attack advantage monotonically increases during standard training across various attacks, which demonstrates the underlying relations between them.
>
> Finally, we clarify that it is not necessary to find a perfect proxy for privacy risk for the motivation. Even though the loss gap might only reflect the privacy risk caused by loss-based attacks, it still leads to the conclusion: Only a small fraction of parameters substantially impacts the privacy risk (at least for loss-based attacks). This is sufficient to motivate our method design, which mitigates the privacy risk via those most important parameters. As a result, the gradient can also reflect the privacy sensitivity with the loss proxy.
>
> **2. The impact of the weight magnitude on the privacy risk [W3]**
>
> We clarify that the utility-privacy trade-offs (shown in Figures 3 and 4) are achieved by tuning the hyperparamters of each method. For our method and the ℓ1 and ℓ2 baselines, the hyperparameter is the magnitude of the weights—$\lambda$. The hyperparameters searched are [0.0004,0.0005,…,0.0014] for the ℓ1 baselines, and [0.6,0.7,0.8,0.9,0.95] for the ℓ2 baselines. For our method, we fix $\lambda=0.001$ and search the $alpha \in [0.3, 1.7]$. Due to the trade-off, the accuracy or the attack advantage will be worse in Figures 3 and 4, if we increase or decrease the $\lambda$ beyond the search ranges. Therefore, the advantage of our method in the trade-offs is solely due to the "privacy-aware".
>
> **3. Clarification of the parameter weight. [W4]**
>
> There might be some misunderstandings. We clarify that $\ell_1$ regularization promotes the **sparsity** in the model parameters, which reduces the number of active parameters (reduce the over-complexity) and then the privacy risk. This is the difference between $\ell_1$ and $\ell_2$ regularizations, and explains why $\ell_1$ outperforms $\ell_2$ in defending privacy attacks. In our method, we focus the sparsity regularization on those parameters that mostly contribute to the privacy risk.

---

> > ### Comment · Reviewer_8dBT · 2024-11-25
> >
> > Thanks for the responses to my comments.
> >
> > I agree with the authors that in practice it might be intractable and unnecessary to find a perfect proxy. However, it is essential that the chosen proxy be reliable.
> >
> >
> > Comment 1: The proof in Appendix E seems to rely on two key assumptions: (1) the attack model is a metric-based attack that uses the loss as the metric, and (2) the losses are distributed according to Gaussian distributions. Consequently, it is unclear whether the conclusion that the loss gap is positively correlated with the attack advantage generalizes to other attack models. If this conclusion does not hold for other attack models, then the loss gap may fail to serve as a reliable proxy for privacy risk under such scenarios.
> >
> > Comment 2: Moreover, while the loss gap may be positively correlated with the attack advantage, this does not, in general, imply that the sensitivities of the parameters based on the loss gap accurately reflect the differences in their impacts on privacy (as measured by the attack advantage). This potential misalignment could lead to suboptimal regularization, where parameters with significant privacy impact are inadequately penalized, reducing the effectiveness of the defense mechanism against membership inference attacks.

---

> > > ### Author Response · Authors · 2024-11-25
> > >
> > > Thank you for the response and new comments. We reply to the comments point by point:
> > > * In Figure 11 of appendix E, we show that the loss gap is consistently and positively related to the attack advantage of various attack methods. The empirical results demonstrate that the loss gap is a reliable proxy for privacy risk.
> > > * As demonstrated above, we use the loss gap as a (approxiated) proxy of the privacy risk. Thus, our analysis utilizes the gradients of the loss gap w.r.t the parameters to analyze the sensitivity as a **practical** method. This work contributes to the community with the **new insight**, validated by the effectiveness of the simple regularization. In future works, the new insight we provided may inspire more specific-designed methods to achieve optimal performance.

---

### Official Review · Reviewer_Nusb · 2024-11-03

**Soundness:** 2
**Presentation:** 3
**Contribution:** 1
**Rating:** 3
**Confidence:** 5

**Summary:**

This paper aims to defend against membership inference attack using sparsity tuning.

The basis intuition is that not all model parameters are equally sensitive in terms of privacy, by empirical analysis the authors found that only a small fraction of parameters are responsible for privacy leakage.   The main idea of mitigation strategy is to prompt sparsity in the model parameter, with the aim of balancing both utility and privacy. Consequently, the sparsity regularization encourages the sensitive parameters to become small, thereby decreasing vulnerability to MIAs.

**Strengths:**

The authors perform a through analysis on the distribution of model parameters and its effects on privacy attacks. It is expected to see that only a small portion of parameters contribute to the privacy  leakage. The following sparsity technique also makes sense.

**Weaknesses:**

While I agree that in general the average loss of members and non-members are different, shrinking  this gap with  the proposed sparsity technique can be effective to those loss-based (or related) MIAs. But it may not be able to defense against  other more advanced state of the art MIAs. Here are my two main concerns:

1. The evaluation metric for privacy is out of dated, this metric has been criticized in many recent MIAs
[R1] D. Hintersdorf, L. Struppek, and K. Kersting, “To trust or not to trust prediction scores for membership inference attacks,” in IJCAI, 2022
[R2] S. Rezaei and X. Liu, “On the difficulty of membership inference attacks,” in IEEE Conf. Comput. Vis. Pattern Recog. (CVPR), 2021
[R3] N. Carlini, S. Chien, M. Nasr, S. Song, A. Terzis, and  F. Tram `er, “Membership inference attacks from first principles,” in IEEE Symposium on Security and Privacy (S&P). IEEE, 2022

I suggest the authors to test the performance using TPR at low FPR rates for assessing the privacy risks.

2. The evaluated MIAs are also out of dated, most of them are metric based MIAs. Their performances are very close to each other and more importantly, these attacks often produce misleading results, which have been extensively explored in
[R4] Li, Xiao, et al. "On the privacy effect of data enhancement via the lens of memorization." IEEE Transactions on Information Forensics and Security (2024).
[R5] Aerni, Michael, Jie Zhang, and Florian Tramèr. "Evaluations of Machine Learning Privacy Defenses are Misleading." arXiv preprint arXiv:2404.17399 (2024).

I suggest the authors to adopt more state of the art MIAs for example LiRA [R6]  and
RMIA [R7] because there are shown to be effective in the TPR at low FPR regions. More importantly, the results of these attacks, particularly LiRA, align consistently with the definition of privacy risk, as well as with the concepts of differential privacy and memorization degree [R4, R5]. This contrasts with the deployed metric-based MIAs, which often yield misleading results by acting as strong non-member detectors. The reason for this discrepancy is that metric-based MIAs typically exploit the fact that models are more confident in members than non-members. However,  this confidence does not hold at a more nuanced level; members with high privacy risk are often difficult and atypical samples, which tend to have low confidence and are therefore frequently misclassified as non-members.
[R6] Carlini, Nicholas, et al. "Membership inference attacks from first principles." 2022 IEEE Symposium on Security and Privacy (SP). IEEE, 2022.
[R7] Sajjad Zarifzadeh, Philippe Liu, and Reza Shokri. Low-cost high-power membership
inference attacks, 2024. URL https://arxiv.org/abs/2312.03262

**Questions:**

How would you expect the trade-off between privacy and utility after adopting new evaluation metric and MIAs?

---

> ### Author Response · Authors · 2024-11-23
> **Response to Reviewer Nusb**
>
> Thank you for the constructive and elaborate feedback. Please find our response below:
>
> **1. Add SOTA shadow-model based attacks [W1]**
>
> Thank you for the suggestion. We add 3 recent SOTA attacks in the revised paper. Please find the detailed reply in the General Response.
>
> **2. Using TPR at low FPR to evaluate [W2]**
>
> Thank you for the great suggestion. We use TPR at low FPR to evaluate the results of the current SOTA attack methods RMIA [1], LiRA [5] and Attack R [4] in Appendix F.3, where PAST still demonstrates strong performance. We will add the results of TPR @ low FPR for other attack methods in the final version.
>
> **3. How would you expect the trade-off after adopting new evaluation metric and MIAs? [Q1]**
>
> Thank you for the question. As present in Appendix F.3, PAST can provide state-of-the-art trade-off under the new attack and evaluation metric.

---

> > ### Comment · Reviewer_Nusb · 2024-11-27
> >
> > Thanks a lot for the response.
> >
> > Thanks a lot for managing to add three recent SOTA attacks and include TPR at low FPR as a metric for evaluating the results.  By inspecting Figure 14 in the revised version, showing the comparison results using CIFAR-10 dataset, I find it is difficult to assess the effectiveness of the proposed approach. If you look at the  bottom plots with TPR at 0.1% FPR, the attack success rate for most cases are nearly zero, this makes it very difficult to assess the performance. I think the performance for the baseline model using Resnet-18 and CIFAR-10 is somewhere is at the scale of 10% TPR at 0.1% FPR, so I donot know why in your case the attach success rate is extremely low, how many shadow models did you use for training?
> >
> > As I said, I am still in doubt of the effectiveness of the proposed approach, therefore I will maintain my score for now.

---

> ### Author Response · Authors · 2024-11-27
>
> Thank you for your concern! Sorry for the confusion, but we would like to clarify that our results are valid and effectively demonstrate the proposed approach's effectiveness.
>
> Regarding the “attack success rate for most cases is nearly zero” in the TPR vs. Test Accuracy curve in Figure 14, we believe this is due to the limited number of inference models and the weakness of attacks except RMIA in this setting (as detailed below). Additionally, we emphasize that attention should be directed to the right-hand region of each plot, as the TPR under higher utility is more meaningful.
>
> As for the baseline should "at the scale of 10% TPR at 0.1% FPR," we clarify that this reflects the privacy level of models without defenses. In Figure 14, all models have applied defenses, so the overall scale should naturally be lower.
>
> Regarding the number of shadow models (or more accurately, inference models, as stated in original paper [1]), we used only 4 inference models due to time constraints. However, based on the results from the RMIA paper [1] (Figure 3), this is already sufficient to approach the best attack performance. Therefore, we believe this is enough to demonstrate the effectiveness of the proposed approach.
>
> [1] Zarifzadeh, et al. Low-Cost High-Power Membership Inference Attacks. ICML 2024.

---

### Official Review · Reviewer_DxZ7 · 2024-11-03

**Soundness:** 3
**Presentation:** 4
**Contribution:** 4
**Rating:** 8
**Confidence:** 4

**Summary:**

The paper introduces Privacy-aware Sparsity Tuning (PAST), a defense mechanism against membership inference attacks (MIAs). The method relies on a key observation that only a small fraction of model parameters significantly contribute to privacy risk, measured by the loss gap between member and non-member data. PAST modifies traditional L1 regularization by applying adaptive penalties to different parameters based on their *privacy sensitivity* - how much they contribute to the member/non-member loss gap. Parameters with higher privacy sensitivity receive stronger regularization, while less sensitive parameters are spared to maintain model utility. The method is implemented as a simple fine-tuning phase after regular training. Authors perform extensive experiments on multiple datasets (Texas100, Purchase100, CIFAR-10/100, ImageNet) and various attack methods. PAST demonstrates superior privacy-utility trade-offs compared to existing defenses, while preserving accuracy. The method is also compatible with and can enhance other existing MIA defenses.

**Strengths:**

I believe it is a very strong paper. It combines an interesting and novel observation (privacy sensitivity is distributed unevenly across model parameters) with a simple yet effective method to utilise this observation to solve a concrete and well-known problem of defending against MIAs. The empirical evidence is also thorough and convincing.

* The paper uses clever technique - computing gradient of the loss gap - to estimate each individual parameter's contribution to the privacy risk. It yields an interesting and novel quantification of per-parameter risk, and shows that only a few parameters need to be regularized to mitigate a privacy risk
* The proposed method (PAST) is a simple yet effective adaptation of L1 regularization, making it easy to implement in practice. Additionally, authors shown computational overhead of adopting PAST to be low compared to the overall training cost.
* The evaluation setup is appropriate and convincing. Specifically, Figures 3 and 4 show the trade-off between attack advantage and model accuracy - which is probably the best way to assess and visualise risks associated with an MIA. Evaluations cover a wide range of MIAs and MIA defences.
* The results show PAST method to be strictly better than other MIA defences and regularization methods in improving privacy-utility trade-off.

**Weaknesses:**

As far as I can see, authors do not consider SOTA shadow-model based attacks like LiRA ([Carlini et al., 2022](https://arxiv.org/abs/2112.03570)) and Attack R ([Ye et al., 2022](https://arxiv.org/abs/2111.09679)). It would be important to understand whether their findings hold for a strong MIA adversary capable of trainings shadow models.

**Questions:**

* How does privacy sensitivity distribution (Fig. 1b) changes throughout PAST training? Similar to the privacy onion effect ([Carlini et al., 2022](https://arxiv.org/abs/2206.10469)), do you think the privacy could "migrate" from parameters you heavily regularize to other parameters?
* Can you clarify the computational aspects of the regularization process?
	* How do you compute $\gamma_i$ in (3)? Do you run an additional forward pass for non-members $\mathcal{S}_n$? If so, an increase of only 10% on the training time is a bit surprising, since you're effectively doubling the size of a batch, and run both forward and backward pass on it.
	* In the equations for $\gamma_i$ does $\nabla_{\theta_i} \tilde{\mathcal{G}_\theta}$ refer to the gradient norm?
* To clarify the experimental setup (4.1, Datasets). Do you use different non-member sets for PAST training and MIA evaluations?

---

> ### Author Response · Authors · 2024-11-23
> **Response to Reviewer DxZ7**
>
> Thanks for your positive and valuable suggestions. Please find our response below:
>
> **1. Additional SOTA attacks [W1]**
>
> Thank you for the suggestion. Please find the detailed results in the General Response.
>
> **2. Change of privacy sensitivity distribution [Q1]**
>
> Thank you for your suggestion. We analyzed the changes in privacy sensitivity during the PAST training process and confirmed the existence of privacy migration in Appendix F.1. Specifically, we plotted the privacy sensitivity distribution at the 1st, 25th, and 50th Epochs during the PAST training, with privacy sensitivity vs. parameter index. We present the results in Figure 12 and show that the distributions indeed migrate among the different epochs.
>
> The migration does not affect the effectiveness of our method as the proposed regularization will be empasized on those "important" parameters in each epoch, which can be dynamic and different. In Figure 13a, we empirically show that the overall privacy sensitivity of model parameters in PAST training is **significantly reduced** compared to the standard training. This demonstrates the robustness of our method in handling privacy risks.
>
> **3. Computational aspects of the regularization process [Q2]**
>
>   * **The computation of $\gamma_i$ in Eq.(3)**
> Yes, we run an additional forward pass for non-members, which indeed takes approximately twice the original computation time. But, our method is designed for tuning, which employs the regularization after the model convergence in the training with the classification loss. Thus, the extra computational cost only incurred during the few tuning epochs (we set as **20** epochs), as described in line 237. That's why our method does not lead to heavy computational cost.
>
>   * **The meaning of $\nabla_{\theta_i}\widetilde{\mathcal{G}}_\theta$**
> Thank you for pointing out the typo. Here, it should be **$\nabla_{\theta_i}\mathcal{G}_\theta$**, which represents the gradient of loss gap $\mathcal{G}_\theta$ with respect to a specific parameter $\theta_i$, as defined at lines 173 and 205. We fix this typo in the revised version.
>
> **4. Details of dataset splitting [Q3]**
>
> Thank you for the question. Yes, we use different non-member sets for PAST training and MIA evaluations. Following previous work [1],  we divide the entire dataset into six subsets: target train, target reference, target test, shadow train, shadow reference, shadow test. We provide the usage of these datasets below.
> * During the PAST training, we use the target reference as non-members for training target models, and use shadow reference as non-members for training shadow models.
> * For MIA evaluations, the shadow test set provides non-members for training the attack model, while the target test set provides non-members for the final evaluation (line 232).
>
> [1] Liu, Z., et al. Mitigating Privacy Risk in Membership Inference by Convex-Concave Loss. ICML 2024.

---

> > ### Comment · Reviewer_DxZ7 · 2024-11-25
> >
> > I appreciate the responses to mine and other reviews. Specifically, I want to thank the authors for including the results for SOTA MIAs and TPR@low FPR as a metric - this was my major concern, echoed by other reviewers.
> >
> > Reading through other reviews, the major potential issue with the paper is that the method, heavily relying on the loss gap as a proxy metric, would only work for simple attacks. I believe the latest update provided my the authors effectively addresses these concerns - having the results on LiRA/RMIA/Attack R is a strong signal that the proposed method is effective for a wide range of attacks. This empirical evidence, in my opinion, also justifies the intuition behind using the loss gap as a proxy.
> >
> > Overall, I think authors have sufficiently addressed concerns raised by the reviewers - and I choose to maintain my high score

---

> > > ### Author Response · Authors · 2024-11-25
> > >
> > > Thank you so much for the recognition. We are pleased that our response addressed your concerns, which also improves the quality of this work.

---

### Author Response · Authors · 2024-11-23
**General Response**

We thank all the reviewers for their time and valuable comments. We are glad that reviewers find this work focuses on an **important topic** in machine learning privacy (5igj) with a **novel** quantification of per-parameter privacy risk (DxZ7). We are also encouraged that reviewers find the method **simple and effective** (DxZ7), supported by **thorough analysis** (Nusb) and **extensive experiments** demonstrating significant improvements (DxZ7, 8dBT). Additionally, reviewers recognize the **clarity** of the paper (5igj) and its **relevance** (8dBT, 5igj). We provide point-by-point responses to all reviewers’ comments and concerns.

In the following responses, we have addressed the reviewers' comments and concerns point by point. The reviews allow us to strengthen our manuscript and the changes$^1$ are summarized below:
* Added new MIA attack methods: **RMIA** [1], **LiRA** [2] and **Attack R** [3]; Added a new evaluation metric in **Appendix F.3** (DxZ7, Nusb, 5igj).
* Included a discussion on PRIVACY "MIGRATION" in **Appendix F.1** (DxZ7).
* Analyzed the rationale and generalizability of the loss gap as a privacy proxy in **Appendix E** (8dBT).
* Added analysis of privacy sensitivity within each layer in **Appendix F.2** (5igj).

Here, we provide a response to the common concern of reviewers.

## Additional results of SOTA attacks:

We note that three reviewers suggested adding state-of-the-art MIA methods or evaluation metrics. Due to the time limit, We add the results of recent attack methods -- **RMIA** [1] (ICML 2024), **LiRA** [2] (S&P 2022) and **Attack R** [3] (CCS 2022) in Appendix F.3. We plotted the AUC-Test accuracy curve and the TPR@0.1%FPR-Test accuracy curve, both of which show that **PAST consistently outperform other defenses**. We will add more results of other SOTA attacks in the final version.

[1] Zarifzadeh, et al. Low-Cost High-Power Membership Inference Attacks. ICML 2024.

[2] Carlini, N., et al. Membership inference attacks from first principles. S&P 2022.

[3] Ye, J., et al. Enhanced membership inference attacks against machine learning models. CCS 2022.

---
$^1$ For clarity, we highlight the revised part of the manuscript in blue color.

---

### Meta-Review · Area_Chair_Ldod · 2024-12-21

**Metareview:**

The submission introduces, PAST, a defense mechanism against Membership Inference Attacks (MIAs). It is based on the observation that only a small fraction of the model parameters contribute significantly to the privacy risk, measured by the loss gap between member and non-members data. The PAST mechanism then modifies the traditional L1 regularization by applying adaptive penalties to different parameters depending on their privacy sensitivity.

The main concern about this paper is that the proposed method lacks theoretical rigor. While the authors added a rationale in the Appendix, the assumptions made require further justification.

**Additional Comments On Reviewer Discussion:**

Different concerns were raised by the reviewers including:
1) Missing baselines
2) Limited metrics
3) Lack of theoretical rigor

While the author(s) satisfactorily addressed 1) and 2) in the discussion, the concern around 3) remains.

---

### Decision · Program_Chairs · 2025-01-22

Reject